# Automatic Text Summarization of Biomedical Text Data: A Systematic Review

**Andrea Chaves** , **Cyrille Kesiku** and **Begonya Garcia-Zapirain** *

eVida Research Group, University of Deusto, Avda/Universidades 24, 48007 Bilbao, Spain
* Correspondence: mbgarciazapi@deusto.es

**Abstract:** In recent years, the evolution of technology has led to an increase in text data obtained from many sources. In the biomedical domain, text information has also evidenced this accelerated growth, and automatic text summarization systems play an essential role in optimizing physicians' time resources and identifying relevant information. In this paper, we present a systematic review in recent research of text summarization for biomedical textual data, focusing mainly on the methods employed, type of input data text, areas of application, and evaluation metrics used to assess systems. The survey was limited to the period between 1st January 2014 and 15th March 2022. The data collected was obtained from WoS, IEEE, and ACM digital libraries, while the search strategies were developed with the help of experts in NLP techniques and previous systematic reviews. The four phases of a systematic review by PRISMA methodology were conducted, and five summarization factors were determined to assess the studies included: *Input*, *Purpose*, *Output*, *Method*, and *Evaluation metric*. Results showed that 3.5% of 801 studies met the inclusion criteria. Moreover, *Single-document*, *Biomedical Literature*, *Generic*, and *Extractive* summarization proved to be the most common approaches employed, while techniques based on *Machine Learning* were performed in 16 studies and *Rouge* (Recall-Oriented Understudy for Gisting Evaluation) was reported as the evaluation metric in 26 studies. This review found that in recent years, more transformer-based methodologies for summarization purposes have been implemented compared to a previous survey. Additionally, there are still some challenges in text summarization in different domains, especially in the biomedical field in terms of demand for further research.

**Keywords:** medical documents; text summarization; language processing; intrinsic evaluation





## 1. Introduction

The accelerated growth of textual data for academia, research, and industry from diverse sources of information such as news, books, journals, scientific articles, databases, and medical records, among others, has become a significant landmark in the further development of different technologies aimed at extracting the most meaningful information in different application domains [1].

The growth of large amounts of textual information has also become evident in the last few years, particularly in the biomedical field [2–5]. Health care professionals face reviewing a huge amount of texts on a daily basis [6,7] due to the meaningful knowledge and information this data includes, such as symptoms, treatment, usage of drugs, and reactions, among others [8]. This information may be represented as biomedical literature, clinical notes, or reports in electronic health records *(EHRs)*, while textual data as biomedical literature refers to text data found in databases of scientific articles and *EHR* includes electronic records of patients' health information [9].

Natural Language Processing *(NLP)* techniques have become indispensable mechanisms for several applications in the biomedical domain, as they allow either illnesses based on narrative medical histories or useful information from clinical notes to be identified or even patient phenotypes from available text resources determined [10–13]. Recently, one

of the most relevant applications has been the search for explicit and implicit knowledge in biomedical literature, since these techniques may provide the basis for better understanding of human diseases and enhance the quality of disease diagnosis, prevention, and therapy [14]. Therefore, NLP-based approaches as a purpose of study provide a basis for improvement in further specific applications that fulfill real-world requirements, such as the extraction of significant information from biomedical textual data. Automatic text summarization is one NLP task method that aims to achieve this goal.

The study of different applications in text summarization has been subject to considerable attention in recent years [15–19], although such applications in biomedical textual data tend to prevail more than in other research fields [20]. Some of the main benefits of using these systems concern the *time saving* and assistance provided to people in the *selection of relevant information*. Automatic text summarization aims to offer a subset of the source text that highlights the most significant points with the minimal possibility of redundancy [15]. Thus, text summarizing reduces data to make it easier for users to find and process important information more quickly and accurately. These methods have tended to be employed in the biomedical area mainly for summarizing research publications or EHR data [21], although data extraction of clinical trial descriptions has also been included in recent research [22–24]. As a result, text summarizing has become a significant tool in helping physicians and researchers organize their information and expertise [21].

In this sense, there is a requirement for working on and exploring existing techniques in the state of the art, mainly for automatic text summarization tasks that focus on problem-solving in biomedical domains where, besides growth in terms of a large amount of existing textual information, professionals also provide crucial information about new health care findings. Thus, to contribute to the development and prevention of health, these studies would in turn contribute to scientific advances that focus on human well-being.

This survey is mainly aimed at identifying the most recent methods, main areas of application, most common evaluation metrics, and types of data, including the most used datasets that are frequently used when text summarization approaches are employed in different biomedical fields. The article is structured as follows. First, we explain how the methods and methodologies were used to develop the systematic review. Then, we provide the results and analysis of the records included. Last, the discussion and conclusion provide recommendations for further research.

## 2. Materials and Methods

The main goal of our systematic review is to identify current challenges that summarization systems have to deal with in order to process biomedical text information. The knowledge found in this study will be used as a starting point for future research in this area. In Table 1, the research questions and objectives that are addressed throughout this study are shown.

**Table 1.** Research questions.

|     | Question | Purpose |
| --- | --- | --- |
| Q1 | What are the most prevalent methods used for text summarization in the biomedical domain? | To determine which techniques have been applied in text summarization in the biomedical domain. |
| Q2 | What data types are used in text summarization in the biomedical domain? | To identify which types of text are most common, either single or multiple document. This will also allow us to assess the most frequently used application in biomedical literature or EHR. |
| Q3 | Which areas in the biomedical field have applied text summarization techniques? | To find out which medical areas have implemented summarization methods. |
| Q4 | What are the most common evaluation metrics of text summarization in the biomedical field? | To assess and identify suitable evaluation metrics to use when comparative studies are carried out on text summarization, mainly in the field of health care. |

The review methodology was based on the *Systematic Review* in the biomedical domain performed by Mishra et al. [25] in conjunction with PRISMA guidelines [26]. In the following subsection, we will cover the steps suggested to identify and screen the studies in our systematic review.

## 2.1. Data Selection

The data collected in this study were obtained from three databases: WoS, IEEE, and ACM digital libraries. The time window was limited from January 1st 2014 to March 15th 2022, with the aim of preventing overlap with past systematic reviews [21,25], while searches mainly focused on text summarization techniques for health care via an engineering approach. The search strategies were developed with the help of experts in Natural Language Processing techniques and the previous systematic review of text summarization in the biomedical domain by Mishra et al. [25]. The screening of records was attached employing the four phases of a systematic review according to PRISMA methodology and are described below.

### 2.1.1. Identification

The main aim was to gather articles that include text summarization terms that focus on the medical domain, with the medical and technical keywords determined in our search strategy being shown in Table 2. The search did not use any population filter—on the contrary, *all fields* were explored. 801 records were found in total in accordance with the search strategy and the three databases proposed.

**Table 2.** Search strategy.

| Medical Keywords | Technical Keywords | Search Strategy |
|---|---|---|
| Biomedical OR biomedicine OR medical OR medicine OR healthcare OR health OR "patient care" OR clinical OR disease OR diseases OR therapy OR therapies OR treatment OR treatment OR diagnosis OR diagnoses OR diagnostic OR etiology | "Text summarization" OR "abstractive summarization" OR "extractive summarization" OR "abstractive text summarization" OR "extractive text summarization" OR "single document summarization" OR "multi-document summarization" OR "query-based summarization" OR "generic summarization" OR "hugging face" | *All fields* (Medical Keywords) AND *All fields* (Technical Keywords) AND (1 January 2014: 15 March 2022) |

### 2.1.2. Screening

The screening process involved applying the first filter with the aim of removing duplicate records. 68 texts were repeated in total among three databases and 733 corresponded to the remaining records.

### 2.1.3. Eligibility

This phase was conducted in order to find original research that included development or evaluation methods in the medical domain capable of summarizing either *biomedical literature* or *Electronic Health Records EHR* documents. Therefore, at this point, the text selection process was carried out in two stages. First, a title review was conducted to exclude studies that were not articles written in English, with opinion papers or their contents not being based on summarization techniques in the biomedical domain. Following this step, 641 papers were discarded, leaving 92 in total remaining.

In the second stage, the abstract screening process focused mainly on being in accordance with eligibility criteria.

1. Studies or summarization tools that describe the evaluation component (metric) and method(s) used.
2. Related Natural Language Processing techniques that can be used as text summarization methods (e.g., text mining, text generation).

In accordance with the aforementioned, 55 texts were excluded following the application of eligibility criteria, resulting in 37 remaining articles to be read completely.

### 2.1.4. Included

According to two previous phases, the inclusion and exclusion criteria for screening records can be defined in Table 3. Following a complete reading of the 37 remaining documents, 28 were then selected for inclusion in the study.

**Table 3.** Inclusion and exclusion criteria.

| Criteria | |
|---|---|
| **Inclusion** | **Exclusion** |
| Complete records. Studies published in journals or at conferences, where the words obtained from the search strategy appear in the title and abstract. Studies that describe the evaluation component (metric) and method(s) used. | Studies not written in English, editorials, or opinion papers. Studies based on summarization techniques in fields other than the biomedical domain. Unavailable records. |

Figure 1 summarizes the whole process of our systematic review in text summarization for the biomedical domain as a flow chart. Here, we can note the different phases suggested by PRISMA methodology.

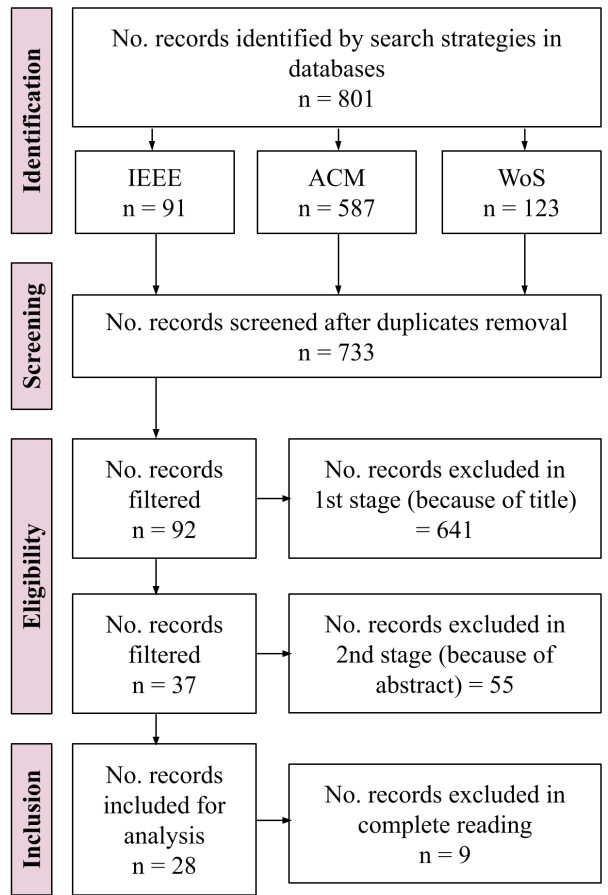

**Figure 1.** Flow chart of systematic review for text summarization in the biomedical domain.

*2.2. Summarization Factors*

In order to evaluate and develop summarization systems, Mani and Maybury in [27] and Jones K. [28] determine that most summarization factors are hard to define and difficult to capture in order to guide the summarization process in particular cases. Therefore, to emphasize the range and varieties in terms of summarizing, *context factors* can mainly be categorized as: input, purpose, and output. Additionally, the method and summary evaluation metrics used are fundamental factors as comparison criteria, since the objectives set out by this review seek to evaluate the most commonly used methods (Q1) and metrics (Q4) when automatic summaries in the biomedical field are generated. As an added factor, the fact of whether an evaluation has been carried out by human experts will also be determined. In Figure 2, the five classifications used for summarization factors are shown and described below.

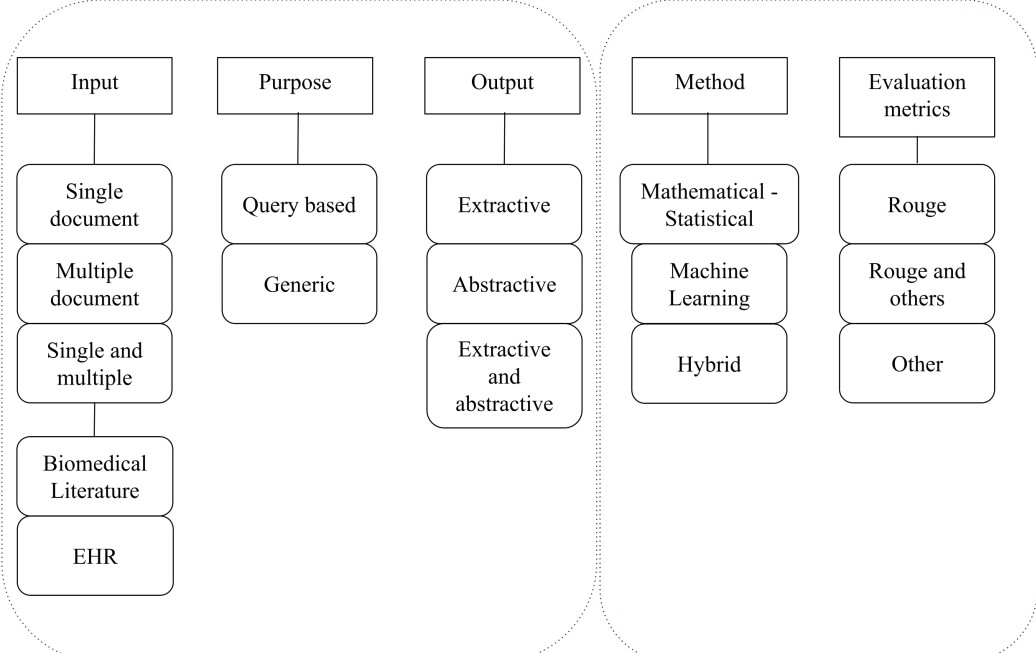

**Figure 2.** Classification of summarization factors.

2.2.1. Input

This category is considered as a *unit input parameter* of *source*, which indicates the number of documents to be summarized according to system, either *single-document* or *multiple-document* [28–30]. Some studies evaluated their proposed summarization systems with single and multiple input texts, hence *single and multiple* classification was also included [31,32]. In addition, as Mishra et al. in [21] proposed, another input category of documents should be taken into account according to the nature of the biomedical field as *Biomedical literature*, which refers to text data found in databases of scientific articles or *EHR* that includes electronic records of patient health-related information (e.g., demographics, progress notes, problems, medication, vital signs, past medical history, immunizations, laboratory data, and radiology reports) [9]. Furthermore, clinical trials were considered as EHR input medical text [33].

2.2.2. Purpose

This criteria concerns the information that a system needs to identify in order to produce a summary. Thus, it is divided into *generic*, where the summary is created based on all the information in document(s), and *query-based*, also known as *user-oriented*, where the summary generated depends on the specific information needed by the user [21,25].

### 2.2.3. Output

Output factors usually refer to assessment of the information included in summaries generated, whether *extracted* from the original input (e.g., sentences, paragraphs, phrases) or an *abstraction*, which generates new text based on the original document with the most salient concepts [15,25]. One of the methodologies also proposes the generation of the summaries based on the two types of output. First, an extractive phase is used to determine the ideas with the highest score according to different ranking steps; then, the summary is composed under the application of an abstractive method [32]. This method was classified as *extractive and abstractive* output category.

### 2.2.4. Method

With a view to determining the approaches used by summarization systems, we have classified the methods into three general categories, *Mathematical-Statistical*, *Machine Learning*, and *Hybrid*. *Statistical* approaches use statistical features of text, usually use sentence ranking based on a mathematical formula that ranks each sentence according to some factors as keywords, term frequency or sentence location, and then the sentence with the best score is selected for inclusion in the summary [21,34]. Therefore, it is said that *Statistical* techniques are based on the *Edmundsonian paradigm* [35]. Graph-based models were included as *mathematical* approaches, providing a text representation that may be used to rank sentences according to significance. Usually, a graph-based summary is based on a single relevance indicator obtained from the centrality of phrases in its graphic representation [15], while the *Machine Learning* category refers to the methods that need to learn the weight of each indicator from a corpus of text data. These techniques provide a great deal of flexibility because the number of important indicators is almost limitless [15]. The different models based on deep learning techniques such as transformers were also included in this category.

### 2.2.5. Evaluation Metrics

The evaluation of summarization system performance is categorized as either *extrinsic* or *intrinsic* [36]. The extrinsic technique assesses the influence of summarizing on the quality of specific activities that rely on summaries generated, such as success rate, time-to-completion, and decision-making accuracy, among others [21,37]. On the other hand, intrinsic metrics assess summarizing skills using measurements that evaluate the quality of summaries (e.g., informativeness, accuracy, relevance, comprehensiveness, readability, precision, recall, F-measure) [37–39]. Since we intend to assess the most common evaluation metrics of summarization systems in the biomedical field (Q4), we will focus on intrinsic evaluation metrics.

## 3. Results

### 3.1. Study Frequency According Geographical Distribution, Years, and Type of Publication

The first stage of eligibility according to title resulted in 92 remaining records, which will subsequently be evaluated based on abstract screening. At this point, a frequency study of articles was carried out, the goal being to categorize the number of studies according to geographical distribution, years, and type of publication. The results are shown in Table 4 and Figure 3. As we can see, most text summarization studies in the biomedical domain have been written in Asia, accounting for 57.61% of the 92 records in the time window selected, with a greater contribution (32.61%) from the south of the continent (Figure 3a). It is also shown that in the period from 2014 to 2022 (March), research in this field was the subject of attention within the scientific community, and the steady increase in the number of articles published over these years is evident in Figure 3b. Last, in Figure 3c, a greater contribution from articles published in conferences than in journals is shown.

**Table 4.** Number and frequency of studies categorized according to geographical distribution, year, and type of publication following first eligibility stage.

| Parameters | Category | Frequency | |
|---|---|---|---|
| | | No. Studies | % |
| **Location** | Eastern Africa | 1 | 1.09% |
| | Northern Africa | 3 | 3.26% |
| | **Africa** | **4** | **4.35%** |
| | Eastern Asia | 22 | 23.91% |
| | Southern Asia | 30 | 32.61% |
| | Western Asia | 1 | 1.09% |
| | **Asia** | **53** | **57.61%** |
| | Northern Europe | 1 | 1.09% |
| | Southern Europe | 4 | 4.35% |
| | Western Europe | 9 | 9.78% |
| | **Europe** | **14** | **15.22%** |
| | North America | 16 | 17.39% |
| | South America | 3 | 3.26% |
| | **America** | **19** | **20.65%** |
| | **Australia/Oceania** | **2** | **2.17%** |
| **Year** | 2014 | 3 | 3.26% |
| | 2015 | 6 | 6.52% |
| | 2016 | 5 | 5.43% |
| | 2017 | 5 | 5.43% |
| | 2018 | 12 | 13.04% |
| | 2019 | 15 | 16.30% |
| | 2020 | 20 | 21.74% |
| | 2021 | 22 | 23.91% |
| | 2022 | 4 | 4.35% |
| **Type of publication** | Conference | 49 | 53.26% |
| | Journal | 43 | 46.74% |

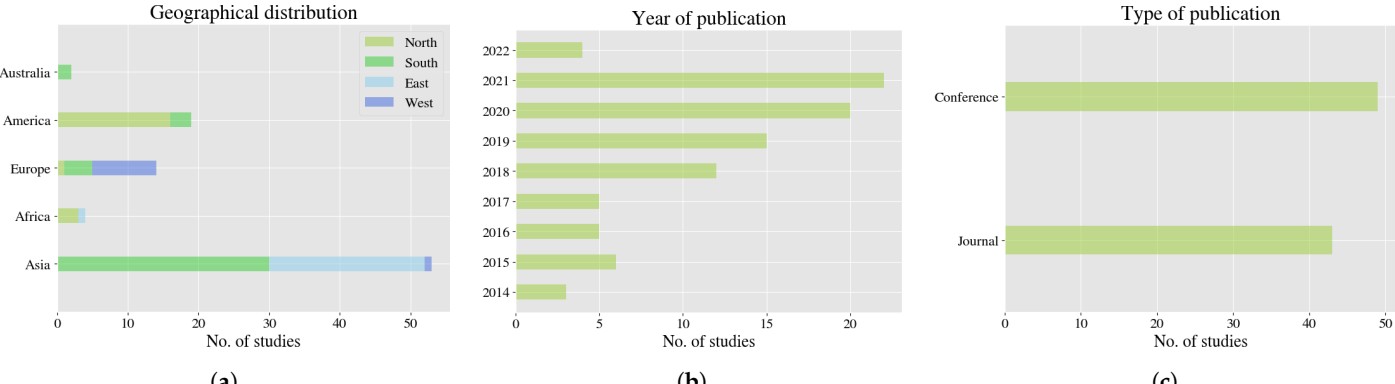

**Figure 3.** Number of studies following first stage of eligibility criteria. (**a**) Geographical distribution. (**b**) Year of publication. (**c**) Type of publication.

### 3.2. Study Frequency according to Summarization Factors

Table 5 and Figure 4 show the number and frequency of studies according to the summarization parameters previously mentioned for 28 records that met inclusion criteria. We can see that the main application is *biomedical literature* and most summarization systems were applied to *single document* text. Only occasionally was summarization of *multidocument* in biomedical literature carried out, and never applied to *EHR* (Figure 4a). The information that systems required to produce summaries is mostly *generic*, with only 10% basing

their summaries on information provided by the user *query-based* (Figure 4b). 75% of the summary system outputs were text *extracted* from data input, 21% were an abstraction generated from the input document, and only one system produced an extractive and abstractive output (Figure 4c). Sixteen studies (57%) generated summaries using *Machine Learning* approaches and, in conjunction with *Mathematical-statistical* models (8; 28%), were the most common summarization methods, with only four (14%) studies using both approaches (Figure 4d). The performance of 92% of systems (26) was evaluated using at least one *Rouge* metric, with hardly any of the 7% of studies evaluated having used a different metric (Figure 4e). Last, only some of the studies (7; 25%) incorporated a human evaluation of the summaries generated (Figure 4f).

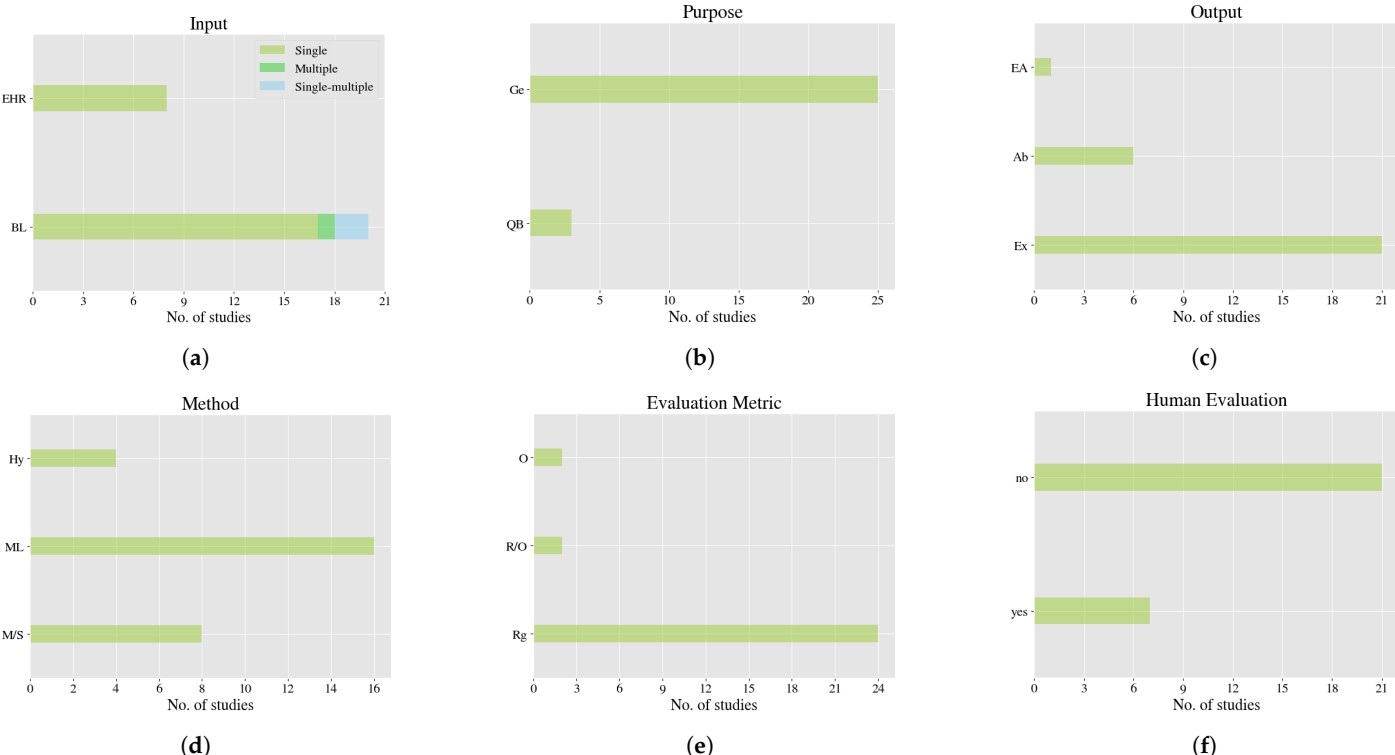

**Figure 4.** Classification of studies following second stage of eligibility criteria according to summarization factors. (**a**) Input. (**b**) Purpose. (**c**) Output. (**d**) Method. (**e**) Evaluation Metric. (**f**) Human Evaluation.

Finally, Table 6 shows the features and descriptions of the five summarization factors for each record that met the inclusion criteria defined in Table 3. The geographical distribution, year, and type of publication of studies are included.

To determine the most relevant studies in this review, the number of citations in Google Scholar to date was evaluated. This evaluation was developed for each type of method used—*Mathematical-Statistical*, *Machine Learning*, and *Hybrid*—and the type of technique (Output) used is also included. The results are shown in Table 7.

**Table 5.** Number and frequency of studies categorized according to summarization factors following second eligibility stage.

| Parameters | Category | Frequency | |
|---|---|---|---|
| | | No. Studies | % |
| **Input** | Single-document (SD) | 25 | 89.29% |
| | Multiple-document (MD) | 1 | 3.57% |
| | Single-multiple-document (SMD) | 2 | 7.14% |
| | Biomedical literature (BL) | 20 | 71.43% |
| | EHR (EHR) | 8 | 28.57% |
| **Purpose** | Query-based (QB) | 3 | 10.71% |
| | Generic (Ge) | 25 | 89.29% |
| **Output** | Extractive (Ex) | 21 | 75.00% |
| | Abstractive (Ab) | 6 | 21.43% |
| | Extractive and abstractive (EA) | 1 | 3.57% |
| **Method** | Mathematical/Statistical (M/S) | 8 | 28.57% |
| | Machine Learning (ML) | 16 | 57.14% |
| | Hybrid (Hy) | 4 | 14.29% |
| **Evaluation Metric** | Rouge (Rg) | 24 | 85.71% |
| | Rouge and others (R/O) | 2 | 7.14% |
| | Other (O) | 2 | 7.14% |
| **Human Evaluation** | Human evaluation (HE) | 7 | 25.00% |
| | No human evaluation (NHE) | 21 | 75.00% |

**Table 6.** Studies included and their characteristics according to summarization factors.

| Title | C/J | Loc. | Year | Input | Purpose | Out | Method (Best) | Metric (Best) | H. Evaluation |
|---|---|---|---|---|---|---|---|---|---|
| Ontology-Aware Clinical Abstractive Summarization [40] | C | USA | 2019 | SD, EHR: Radiology Reports | Ge | Ab | ML: pointer–generator based on Seq2Seq model | Rg 1:38.42 2:23.29 L:37.02 | HE: Radiologist (Readability, Accuracy, Completeness) |
| Extractive Text Summarization using Ontology and Graph-Based Method [41] | C | Singapore | 2019 | SD, BL: Review papers | Ge | Ex | M/S: Graph-based method (PageRank) | Rg-P 1:25.46 L:23.61 | NHE |
| Domain-Aware Abstractive Text Summarization for Medical Documents [42] | C | Spain | 2019 | SD, BL: abstracts from PubMed dataset | Ge | Ab | ML: deep-reinforced pointer–generator network | R/O- 1:42.43 2:21.59 L:36.89 TFIDF UMLS MeSH | NHE |
| Knowledge-Infused Abstractive Summarization of Clinical Diagnostic Interviews: Framework Development Study [43] | J | USA | 2021 | SD, EHR: Diagnostic interviews by mental health professionals | QB: clinical diagnostic interviews | Ab | M/S: knowledge-infused abstractive summarization (KiAS) | Rg-L R:24.46 F1:32.57 | HE: Mental Health professionals (GQCC, GQUC, meaningful responses) |
| Extractive summarization of clinical trial descriptions [22] | J | Germany | 2019 | SD, EHR: clinical trial descriptions from clinicaltrials.gov | Ge | Ex | ML: TextRank | Rg-L P:30.95 R:33.86 F1:30.03 | HE: Human reviewers (Contains all information, Helpfulness) |
| Biomedical-domain pretrained language model for extractive summarization [44] | J | China | 2020 | SD, BL: titles and abstracts from PubMed dataset (Task 6a) | Ge | Ex | ML: domain-aware bidirectional language model (BioBERTSum) | Rg-F1 1:37.45 2:17.59 L:29.58 | NHE |
| Deep contextualized embeddings for quantifying the informative content in biomedical text summarization [45] | J | Austria | 2020 | SD, BL: articles from BioMed Central database | Ge | Ex | Hy: deep bidirectional language model and clustering method (BERT-based, BERT-large) | Rg 1:75.04 2:33.12 | NHE |
| CERC: an interactive content extraction, recognition, and construction tool for clinical and biomedical text [46] | J | England | 2020 | SD, BL: abstracts from Medline | Ge | Ex | ML: multistage algorithm (MINTS) | Rg 1:41.4 2:13.6 SU4:17.1 | NHE |

**Table 6.** *Cont.*

| Title | C/J | Loc. | Year | Input | Purpose | Out | Method (Best) | Metric (Best) | H. Evaluation |
|---|---|---|---|---|---|---|---|---|---|
| Evolutionary Algorithm based Ensemble Extractive Summarization for Developing Smart Medical System [6] | J | India | 2021 | SD, BL: PubMed and MEDLINE journal citations | Ge | Ex | Hy: Multiobjective Evolutionary Algorithm based on Decomposition (MOEAD) | Rg-F1 1:70.7 2:65.5 SU:47.9 | NHE |
| Different approaches for identifying important concepts in probabilistic biomedical text summarization [7] | J | Iran | 2018 | SD, BL: Biomedical articles | Ge | Ex | M/S: Bayesian method | Rg 1:78.86 2:35.29 SU4:41.04 | NHE |
| CIBS: A biomedical text summarizer using topic-based sentence clustering [31] | J | Iran | 2018 | SMD, BL: abstracts from PubMed and BioMed | Ge | Ex | ML: Clustering and Itemset mining (CIBs) | Rg 2:34.75 SU4:39.78 | NHE |
| Modified Bidirectional Encoder Representations From Transformers Extractive Summarization Model for Hospital Information Systems Based on Character-Level Tokens (AlphaBERT): Development and Performance Evaluation [47] | J | Taiwan | 2020 | SD, EHR: diagnoses from National Taiwan University Hospital | Ge | Ex | ML: BERT-based structure with a two-stage training method (AlphaBERT) | Rg 1:76.9 2:61.0 L:75.1 | HE: Doctor feedback (Score) |
| Summarization of biomedical articles using domain-specific word embeddings and graph ranking [48] | J | Austria | 2020 | SD, BL: articles from PubMed | Ge | Ex | Hy: domain-specific word embedding and graph-based model | Rg 1:76.87 2:34.91 | NHE |
| MultiGBS: A multilayer graph approach to biomedical summarization [49] | J | Iran | 2021 | SD, BL: articles from BioMed Central | Ge | Ex | M/S: graph-based creation and sentence selection model (MultiGBS) | Rg/O Rg-F1 1:16.4 2:05.2 L:14.6 SU4:07.5 Bertscore F1:80.6 | NHE |

| Title | C/J | Loc. | Year | Input | Purpose | Out | Method (Best) | Metric (Best) | H. Evaluation |
|---|---|---|---|---|---|---|---|---|---|
| Quantifying the informativeness for biomedical literature summarization: An itemset mining method [38] | J | Iran | 2017 | SD, BL: Scientific papers | Ge | Ex | M/S: Itemset mining | Rg 1:75.83 2:33.81 SU4:38.89 | NHE |
| Frequent itemsets as meaningful events in graphs for summarizing biomedical texts [50] | C | Iran | 2018 | SD, BL: scientific articles | Ge | Ex | M/S: Graph-based method | Rg 2:34.03 SU4:38.51 | NHE |
| Nutri-bullets: Summarizing Health Studies by Composing Segments [51] | C | USA | 2021 | MD, BL: scientific abstracts from PubMed and ScienceDirect | Ge | Ab | ML: reinforcement learning (Blank Language Model—BLM) | O-Meteor Me:15.0 | HE: (Faithfulness, Relevance, Fluency) |
| Self-supervised extractive text summarization for biomedical literature [52] | C | USA | 2021 | SD, BL: Radiation Therapy scientific articles from PubMed | Ge | Ex | ML: BERT | Rg-R 1:71.00 2:59.00 | NHE |
| A Hybrid Multianswer Summarization Model for the Biomedical Question-Answering System [32] | C | Vietnam | 2021 | SMD, BL: Medical Question-Answer Summarization dataset (MEDIQA-AnS) | QB: Question-driven filtering phase | EA | ML: Denoising autoencoder and BART (Extractive Abstractive hybrid model - EAHS) | Rg-F1 1:30.00 2:22.00 L:25.00 | NHE |
| Towards neural abstractive clinical trial text summarization with sequence to sequence models [23] | C | Kenya | 2019 | SD, EHR: clinical trial descriptions from clinical trials.gov | Ge | Ab | ML: Seq2Seq model with attention | Rg-F1 1:40.4 2:15.0 L:33.8 | NHE |
| Extractive Text Summarization for COVID-19 Medical Records [53] | C | India | 2021 | SD, BL: COVID-19 research articles from PubMed, Microsoft Academic and WHO COVID-19 | Ge | Ex | ML: Generative Pre-Trained Transformer 2 (GPT-2) | Rg-F1 1:78.22 2:71.17 L:78.22 | NHE |

**Table 6.** *Cont.*

| Title | C/J | Loc. | Year | Input | Purpose | Out | Method (Best) | Metric (Best) | H. Evaluation |
|---|---|---|---|---|---|---|---|---|---|
| Fine-tuning the BERTSUMEXT model for Clinical Report Summarization [54] | C | India | 2020 | SD, EHR: clinical report summarization dataset | Ge | Ex | ML: Fine-tuned BERTSUMTEXT | Rg-F1 1:50.07 2:39.85 L:49.59 | HE: Doctor's opinion |
| A Hybrid Text Classification and Language Generation Model for Automated Summarization of Dutch Breast Cancer Radiology Reports [55] | C | Netherlands | 2020 | SD, EHR: Dutch breast cancer radiology reports | Ge | Ab | ML: encoder–decoder attention model (EDA) | Rg-F1 1:54.0 2:38.8 L:51.5 | HE: Radiologists (correctness, relevance, comprehensible) |
| Query Specific Focused Summarization of Biomedical Journal Articles [56] | C | India | 2021 | SD, BL: articles from COVID-19 Open Research Dataset (CORD-19) | QB: User required information | Ex | M/S: Optimization and contextual method | Rg 1:47.61 2:19.62 L:44.74 | NHE |
| Exploring Multi-Feature Optimization for Summarizing Clinical Trial Descriptions [24] | C | India | 2020 | SD, EHR: Clinical Trial Descriptions from Mendeley datasets | Ge | Ex | M/S: Multi Feature Optimization (MFO) | Rg-R 1:70.0 2:39.0 L:50.0 | NHE |
| Automatic Text Summarization using Maximum Marginal Relevance for Health Ethics Protocol Document in Bahasa [57] | C | Indonesia | 2021 | SD, BL: Health research ethics protocol | Ge | Ex | M/S: Maximum Marginal Relevance (MMR) | Rg-4 P:34.0 R:71.0 F1:46.0 | NHE |
| Finding Clinical Knowledge from MEDLINE Abstracts by Text Summarization Technique [58] | C | Thailand | 2018 | SD, BL: cervical cancer in clinical trials from MEDLINE abstracts | Ge | Ex | ML: BM25 term-weighting and text filtering techniques | O P:100.0 R:84.0 F1:91.0 | NHE |
| Combining clustering and frequent item set mining to enhance biomedical text summarization [59] | J | USA | 2019 | SD, BL: articles from BioMed central database | Ge | Ex | Hy: clustering and frequent itemset meaning | Rg 1:23.84 2:08.71 SU4:11.45 | NHE |

**Table 7.** Most cited algorithms for each method.

| Method | Algorithm | Output |
|---|---|---|
| *Mathematical-Statistical* | **Bayesian** [7] | *Extractive* |
| *Machine Learning* | Pointer–generator network [40] | *Abstractive* |
| *Hybrid* | Deep bidirectional language model and clustering (BERT-based, BERT-large) [45] | *Extractive* |

## 4. Discussion

As previously stated, due to the accelerated growth of biomedical text data from diverse sources of information, the study of summarization techniques has been subject to considerable attention in recent years [15–18,60]. That is the reason why summarization techniques play a crucial role in this domain and some systematic reviews have been carried out [21,25,55,61]. In this review, we mainly focus on identifying five main parameters in recent summarization systems, which will be discussed below.

Regarding the *Input* factor (question Q2), both *single* and *multiple document* summarization methods have been researched. As the systematic review by Afantenos et al. [25] explains, the notion of using *extractive* methods mainly for *single-document* summarization remains. On the other hand, *multiple-document* summarization is based on *abstractive* techniques [32,51], and aside from the biomedical field, this is also evident in other fields [62–65]. Unlike the review conducted by Mishra et al. [21], the study of techniques in *single-document* summarization has been subject to more attention in the last few years. Moreover, most studies concern summarizing *biomedical literature*. This may be due to the following factors: first, the exponential growth in recent years of scientific literature published in different databases, and second, the ease of access to this information in contrast to accessibility of patient clinical records *EHR*, as we can see in Table A1; among the most common resources used by the studies included in this review, we found seven *open access* datasets for *biomedical literature*, versus two for *EHR*. In addition, there is a great advantage in using biomedical literature as input text, since abstracts or titles of records can serve as ground truth summaries [42,44], resulting in PubMed and BioMed being the most common dataset used for NLP tasks in the biomedical environment (Table A1). Most summarization systems in which *EHR* have been employed as input have tended to base their techniques on *machine learning* approaches [22,23,40,54,55]. In terms of *purpose* factors, although most methods deal with *generic* approaches (89%), some (≈11%) require information taken from different user sources such as clinicians, researchers, and patients [32,43,56].

Regarding question Q4, the performance of systems was mostly evaluated using *intrinsic* and automatic metrics as *(Rouge)*. This is one of the most widely used metrics for summarization tasks and involves counting the number of matching $n-$grams between ground truth and model summaries [66]. The rouge-based evaluation metrics most commonly used by the studies were *Rouge-N*, *Rouge-L*, and *Rouge-SU* of which the *Rouge-L F1 measure* was the most widely used, perhaps due to the correlation existing between this and the evaluation carried out by humans [22,66]. Nevertheless, it is possible to use other evaluation metrics based on biomedical expert tools, such as the UMLS Metathesaurus and MeSH, introduced in [42]. Furthermore, depending on how the summarization problem is presented, it is possible to use classification-based metrics, since selecting relevant sentences is represented as a binary classification issue with all sentences in the input divided into summary and nonsummary sentences [15,58,67]. Although *Rouge* is the most commonly used metric for evaluating summary systems, there is still a gap in determining which one is the most appropriate.

Additionally, there is also a limitation in determining an evaluation that is not based on word matching but on context, since the use of synonyms in the construction of abstracts is one of the main problems when using this type of evaluation metric. Therefore, it is evident that the study of context-based summarization evaluation techniques should be considered. Currently, this problem is starting to be addressed, using metrics based on embeddings. It would be relevant to the incursion of this approach in the biomedical field, since the

semantic understanding of clinical text is a challenge in NLP tasks. This because analyzing clinical unstructured text presents a grammatical challenge in itself [8], the material is often illegible due to the few full sentences, the absence of entire phrases, a high usage of acronyms and abbreviations, and the use of confusing terms [68]. Thus, evaluation metrics that focus on these particular cases would be valuable for the development of the NLP technique in this field. Similarly, another weakness that the evaluation metrics face is the type of summarization technique that the systems use is not taken into account, whether *abstractive* or *extractive*; although the techniques have the same purpose, a summary generated using methods that extract information from the text will be completely different from one based on abstract methods. For this reason, the question arises as to whether rouge-based evaluations are the most appropriate for evaluating abstract or extractive generated texts or, in general, if they are suitable for evaluating any generated text by summarization systems. This limitation opens the way for future studies along these lines. Last, quality evaluations are carried out with the aim of ensuring a broader understanding of how well the system is working. Unlike automatic evaluation, system output is not compared to the reference; rather, in this case, an expert human reviewer manually scores the summary based on its quality [37]. According to Table 6, *human evaluation* was carried out by experts such as radiologists [40,55], mental health professionals [43], doctors [47,54], and also human reviewers [22,51].

Concerning Q1, according to *Rouge L* and *Rouge F1-L* scores, the highest metrics were obtained for extractive-based systems, although in the case of abstractive approaches, the best metrics were obtained using *Transformers-based* models as pointer–generator network [40,42] and encoder–decoder–attention [55]. The above may support the previous research conducted in [69], where it was determined that pointer–generator models improve *Rouge* scores and the fluidity of summaries, especially those based on abstractive techniques. Regarding extractive techniques, the methods that performed positively were AlphaBERT [47] and GPT2 [53], obtaining scores of 75.1 and 78.22 in *Rouge L* and *Rouge F1-L* respectively. Once again, transformer-based methodologies obtained the highest scores in evaluation metrics among the studies in which these values were reported. It is worth noting that the report that used a Bayesian summarization method obtained the highest score according to *Rouge-1*. In general, as has already been demonstrated among other NLP tasks, transformers have been shown to exceed the prediction accuracy of Recurrent Neural Networks (RNNs) and become the industry standard for NLP applications [70]. In particular for biomedical summarization approaches for the period between January 2014 and March 2022, transformers received greater study attention on the part of the scientific community due to their success in performing satisfactorily. With the accelerated growth in the development of methodologies based on *transfer learning*, recent models have offered the development of novel architecture such as *transformers*, which are used to handle sequential input data such as natural language. Unlike RNNs, transformers process the full input at once, and any place in the input sequence is given context by the attention mechanism. This allows for greater parallelization than RNNs, resulting in faster training. A transformer can capture the relationships between words regardless of their position, and so the sequential aspect is not important any more. Therefore, these models can help overcome challenges faced by models that deal with natural language such as long text documents, gradient vanishing, larger training steps, and sequential computation, among others [70].

It is worth noting that the Bayesian statistical method that develops an extractive summarization technique had the highest number of citations to date [7], with respect to all the studies included in the review. For methods based on the machine learning approach, the pointer–generator network presented the highest number of citations, developing an abstractive technique [40], and concerning the hybrid approach, the combination of the BERT transformer-based model together with a clustering technique has been the most relevant [45]. Under this criterion, these summarization techniques applied to biomedical text data have greater recognition, which would indicate the importance of their proposed

techniques. Therefore, future studies that are interested in determining the method of summarization could begin by investigating these proposals that have shown a good level of approval. In short, these models are the most suitable and prevalent of the summarization techniques used to date.

Last, in relation to question Q3, about 71% of studies reviewed developed text summarization based on *Biomedical Literature*, most of them using scientific articles including the title, abstract, and/or corpus of the document from PubMed or Biomed Central datasets (Table A1). Among these studies, the biomedical fields that have been subject to study in the application of automatic summarization techniques in recent years are: *nutrition* [51], *radiation therapy* [52], *COVID-19* [53,56], *cervical cancer* [58], and even input texts concerned with *health research ethics protocol* [57]. Other methods (≈29%) that use *EHR* as *Input* in systems have tended to focus on three main categories: *general radiology reports* [40], including those aimed at breast cancer [55]; *clinical trial descriptions* [22–24]; and *general clinical diagnostics* [47,54] with applications in *Mental Health* [43].

In this sense, summarization systems have been facing some challenges not only in the biomedical field but also in different domains that should lead to future research in this area. Text mining scientists face evident challenges, mainly in the biomedical domain. With regard to performance evaluation, 93% of the studies included used a metric based on Gisting Evaluation (ROUGE), which would indicate that a common metric has been established to evaluate systems. Another issue that summarization systems have been facing is the length of input texts, as we previously discussed—some of the studies included in this review involve transformer-based techniques. Owing to the advantages of using these models, we could say that this problem is being addressed and solutions are emerging, although this issue is not completely resolved. As for the quality of the input information for the purpose of training and testing systems, studies are more inclined to use biomedical literature due to the ease of access to available information and the data structure where, for example, scientific articles provide both text input for the systems and the references as abstracts needed to evaluate their performance. Furthermore, other methods were developed using the title as reference [44]. In contrast, when using clinical records, summarization systems are facing greater challenges that require further and constant study, as they usually evidence many gaps in their phrases, leading to texts with linguistic mistakes, and new terms and acronyms also emerge over time in the biomedical domain [71]. In addition, in the present review, the lack of biomedical text resources in other languages is clearly noticeable. For this reason, future guidelines should focus on the gathering and divulgation of biomedical texts in different languages, since there is also a large amount of valuable information that would also allow the inclusion and evaluation of the different summarization techniques. In short, more research is required to allow and provide publicly accessible summarizing corpora and reference standards in order to assist in the development of summarization technologies in a variety of applications, especially in biomedical text data such as EHR.

## 5. Conclusions

In this study, we present a systematic review of current literature on medical text summarization. The review found that systems mainly focus on *Single Document* and *Biomedical Literature* with a *generic* purpose and *extractive* approach. Among the methods used in relation to *Machine Learning*, *Transfer Learning* methodologies based on transformers have been subject to increased interest in research in recent years compared to previous surveys, obtaining positive results; additionally, *graph-based* models as a representation of textual information and *statistics-based* techniques remain subject to study for the purpose of biomedical text summarization.

Several issues are being addressed, although it is essential to continue researching into possible ways of providing huge improvements, since we can still consider the technology to be a work in progress. Due to the variety of evaluation methodologies and metrics, a meta-analysis still has its limitations, and the absence of common-standard evaluation

methodologies could be a sign of the field's immaturity in comparison to other similar fields in NLP. Defining a favorable method to evaluate the task of summarization is still subject to study; therefore, different methodologies have emerged that, compared to gisting evaluation or $n-$grams, are based on, for instance, word representations of texts in a vectorial form, whereby words that are close in the vector space are likely to have comparable meanings. This evaluation is known as *BERTScore*, and the token similarity is estimated [72,73] using contextual embeddings. This methodology is being used to evaluate the performance of different natural language systems, and so would therefore be of interest when exploring the embeddings in biomedical environments with a view to performing a more specific analysis and assessment. In summarization systems, it would be of great interest to use the vectorial word representation in a particular domain in certain biomedical areas, or also with the different types of input text (EHR or biomedical literature).

Other necessary factors and trends that could have been a focus of attention during the review may have been missing, owing to the fact that we focused on five main features, although as we specified before, there are more summarization factors that could provide essential information about system performance. In addition, we consider that some summarization systems may not have been assessed, this being due to the exclusion criteria we used that involved eliminating articles not authored in English. Nevertheless, the studies included served to achieve the proposed objectives in this review.

**Author Contributions:** Conceptualization, A.C., C.K. and B.G.-Z.; methodology, A.C. and C.K.; formal analysis, A.C.; investigation, A.C.; writing—original draft preparation, A.C.; writing—review and editing, A.C., C.K. and B.G.-Z.; supervision, B.G.-Z. All authors have read and agreed to the published version of the manuscript.

**Funding:** This research received no external funding.

**Institutional Review Board Statement:** Not applicable.

**Informed Consent Statement:** Not applicable.

**Data Availability Statement:** Not applicable.

**Acknowledgments:** The authors would like to thank Mario Jojoa for his help and insights in the search strategy and methodology of this review. The authors also express their gratitude to the eVida Research Group from the University of Deusto, recognised by the Basque Government in accordance with code IT1536-22, for their commitment in providing us with the resources necessary to complete the study.

**Conflicts of Interest:** The authors declare no conflict of interest.

## Appendix A

**Table A1.** Datasets of biomedical resources for summarization tasks.

| Resource | Type of Input Text | Description |
|---|---|---|
| PubMed Central (PMC) | Biomedical Literature | More than 7 million full-text records of biomedical and life sciences journal literature at the U.S. National Institutes of Health's National Library of Medicine (NIH/NLM). *Open access* [74] |
| CRAFT: The Colorado Richly Annotated Full Text Corpus | Biomedical Literature | It is a manually annotated corpus consisting of 67 full-text biomedical journal articles. Each article is a member of the PMC subset. *Open access* [75,76] |
| BioASQ Task-6a | Biomedical Literature | Contains 13 million citations from PubMed dataset, and each citation contains the title and abstract. *Open access* [77] |
| PubMed | Biomedical Literature | Contains more than 34 million citations and abstracts supporting the search and retrieval of biomedical and life sciences literature. *Open access* [78] |

**Table A1.** *Cont.*

| Resource | Type of Input Text | Description |
| --- | --- | --- |
| BioMed Central (BMC) | Biomedical Literature | 300 peer-reviewed journals in science, technology, engineering, and medicine. *Open access* [79] |
| MEDLINE | Biomedical Literature | This database contains more than 29 million references to journal articles in life sciences with a concentration on biomedicine. The records are indexed with NLM Medical Subject Headings (MeSH). *Open access* [80,81] |
| MEDIQA-AnS | Biomedical Literature | The dataset includes 156 questions with related documents as the answers for each. Each answer also has an extractive and an abstractive single-answer summaries and multidocument extractive and abstractive summary considering the information presented in all of the answers. [82] |
| CORD-19: The Covid-19 Open Research Dataset | Biomedical Literature | It is a resource of scientific papers on COVID-19 and related historical coronavirus research. *Open access* [83] |
| Radiology Reports | EHR | 41,066 real-world radiology reports from MedStar Georgetown University Hospital. Each report describes clinical findings about a specific diagnostic case, and an impression summary [40] |
| DIAC-WoZ dataset | EHR | Clinical interviews designed to support the diagnosis of psychological distress conditions created by the Institute for Creative Technologies at the University of Southern California. *Open access* [84,85] |
| NTUH-iMD | EHR | The corpus contains 258,050 discharge diagnoses obtained from the National Taiwan University Hospital Integrated Medical Database and the highlighted extractive summaries written by experienced doctors [47] |
| Clinical trials | EHR | Dataset generation of 101,016 records usable for the summarization task from clinical trials. *Open access* [86] |

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
