# Peer review of "Automatic Text Summarization of Biomedical Text Data: A Systematic Review"

_information, doi:10.3390/info13080393_

Round 1
Reviewer 1 Report
In this work, the authors present a systematic review of automatic text summarization from biomedical text. The review focuses on five summarization factors, including input, purpose, output, method, and evaluation metrics. Overall, the review is comprehensive and well-written. However, there are also some issues that need to be addressed:
1. In Figure 2, the input can be classified as “single document”, “multiple document”, and “single and multiple”. But in section 2.2.1, the type of “single and multiple” is not mentioned. The authors should add some descriptions about this type. In section 2.2.3, the type of “extractive and abstractive” should also be mentioned.
2. Authors may like to add a section to summarize the widely used datasets for the text summarization task, as the dataset is critical for the development and evaluation of automatic text summarization methods.
3. In Table 7, are the rouge metrics from the same dataset? If not, it is not a fair comparison of the different methods.
4. The full name of “Rouge” should be provided at its first occurrence in the main text.
5. Page 1 line 34, “Natural Language processing (NLP)” -> “Natural Language Processing (NLP)”
Reviewer 2 Report
The respected authors presented their work in "Automatic text summarization of biomedical text data: a systematic review". Though a review in nature is not novel, however, this research could use indeed such a review and it would be very valuable.
The manuscript needs to address a aspects few before it can be ready for publications. The most important and significant one is that it corresponds to the inclusion/exclusion criteria assessments and four factors of (input, output, purpose, and methods, and evaluation metric). The only one factor that is relevant is the methods. A survey paper that produces a handful of publications has clearly missed the point.
I highly recommend that both the criteria of inclusion is relaxed so a lot of relevant papers are included. I also recommend that the new set of publications are clustered based on the methods used. Even better, once grouped by the most significant methods, the esteemed authors should select on algorithm (perhaps the most cited algorithm) to summaries all publications in all groups. The result of such an exercise should be compared and reported to the reader. Such an exercise will make this manuscript a powerful tool and present a new era for survey papers making them more exciting to read. There must be a prize for the reader to read a survey paper and this analysis would make the work of the respected authors more meaningful.
The manuscript has suffered minor issues such as using upside-down question marks. Also, the final results in Table 6 if it is still relevant after revisions, it can be pushed into an appendix.

Reviewer 3 Report
The subject of the paper “Automatic text summarization of biomedical text data: a systematic review” is timely and valuable to the audience of the Information Journal. Researchers presented results from a systematic literature review with the use of PRISMA methodology on automatic text data of biomedical text data.
Overall, the paper is well structured, reads quite well, and covers the existing literature quite well. The analysis of the data is interesting and well-documented. However, in my view, some minor amendments are required prior to publication.
I'm missing in the discussion section a part of discovering new knowledge from these 28 studies. The study looks pretty much like a report and does not unveils or discover any significant pattern or trend in those 28 studies that may be worth publishing. The main idea of a literature review is not just to provide a summary of what other studies did but try to find out or present those findings from a new perspective that can show to the reader an existing gap that deserves attention for future research.
Minor comment: Table 4 has wrong values in rows Conference and Journal ((15 and 13), where in Figure 3c are different values.
Round 2
Reviewer 3 Report
Thank you very much. All of my previous comments were correctly addressed. Thank you very much for better claryfing the existing gaps and future guidelines. I think that the manuscript has been significantly improved. I wish you good luck in your future work.